# Using Nutritional Geometry to Explore How Social Insects Navigate Nutritional Landscapes

**DOI:** 10.3390/insects11010053

**Published:** 2020-01-15

**Authors:** Antonin J. J. Crumière, Calum J. Stephenson, Manuel Nagel, Jonathan Z. Shik

**Affiliations:** 1Section for Ecology and Evolution, Department of Biology, University of Copenhagen, Universitetsparken 15, 2100 Copenhagen, Denmark; 2Smithsonian Tropical Research Institute, Apartado Postal 0843-03092, Balboa, Ancon, Panama

**Keywords:** nutritional geometry, cognition, insects, ants, fundamental and realized niches, mutualism

## Abstract

Insects face many cognitive challenges as they navigate nutritional landscapes that comprise their foraging environments with potential food items. The emerging field of nutritional geometry (NG) can help visualize these challenges, as well as the foraging solutions exhibited by insects. Social insect species must also make these decisions while integrating social information (e.g., provisioning kin) and/or offsetting nutrients provisioned to, or received from unrelated mutualists. In this review, we extend the logic of NG to make predictions about how cognitive challenges ramify across these social dimensions. Focusing on ants, we outline NG predictions in terms of fundamental and realized nutritional niches, considering when ants interact with related nestmates and unrelated bacterial, fungal, plant, and insect mutualists. The nutritional landscape framework we propose provides new avenues for hypothesis testing and for integrating cognition research with broader eco-evolutionary principles.

## 1. Introduction

Most insects have sensory traits to select among the many foods that comprise complex nutritional landscapes within natural environments. A central challenge is to harvest a realized nutritional niche (RNN) based on available resources (e.g., in the presence of predation and competition) that optimally targets fitness-maximizing dimensions of the innate fundamental nutritional niche (FNN). Insects also often engage in diverse nutritional mutualisms with bacteria, fungi, plants, and other insects, and we propose that these nutritional subsides can complicate the cognitive challenges of harvesting an optimal RNN. In this review, we focus on ants and use the logic of nutritional geometry (NG) to explore: (1) the neurophysiological traits that insects need to navigate complex nutritional landscapes, and how (2) social environments and (3) mutualist-derived nutrients mediate these foraging decisions.

## 2. Cognitive Mechanisms Governing Nutritional Decisions

Imagine a fruit fly foraging across all the exposed counter-top foods in a kitchen before selecting a single banana on which to oviposit. After navigating a nutritional landscape, the fly has decided that this fruit most closely satisfies the innate nutritional, chemical, and structural requirements of itself and its developing larvae [1]. This fly manages a nutritional life history tradeoff between maximizing its own lifespan by consuming strongly carbohydrate-biased food, or its lifetime egg production by consuming a weakly carbohydrate-biased food [2]. Such behavioral decisions are typically mediated by a simple feedback loop, where nutrient scarcity generates a feeling of hunger that increases ingestion of targeted foods that redress deficient nutrients (Figure 1) [3]. Insects often rely on olfaction to provide the first sensory input used to identify, evaluate and locate optimal food sources (Figure 1) [4]. Gustation is then used to integrate qualitative information about taste commonly divided between bitter, sweet, salty, sour and umami tastes [5,6]. The information from those peripheral senses is then integrated in the higher brain center which enables different forms of learning and memory based on reward/punishment experience (Figure 1) [7].

However, while olfaction and gustation are primed to detect certain compounds in the environment (e.g., harmful substances like toxins or pathogens), the specific nutritional value of foods is rarely encoded by those signals [4,5,6]. This is because peripheral senses provide a rough classification about the food’s expected nutritional content (carbohydrates, micronutrients, and amino acids) [4,5,6]. Instead, internal mechanisms involving metabolic pathways, energy sensors, neurons and neuropeptides evaluate individual metabolic needs and the nutritional value of ingested food (Figure 1) [8]. This provides a feedback with additional information, such as whether the food is harmful or suited for the organism’s metabolic needs (e.g., sweeteners that elicit the perception of sweetness, but cannot be used in energy cycles like sugars) [7]. The nutritional specificity of neuronal networks underlying foraging and feeding behavior has been widely studied in *Drosophila*. Here, experiments manipulating single food nutrients (e.g., carbohydrates, proteins, salts) have been used to describe associated behavioral and physiological responses as well as the neuropeptides and neuromodulators of underlying regulatory processes (reviewed in [8,9]).

Behavioral experiments have shown that insects generally forage to reach performance-maximizing dietary intake targets for specific ratios and amounts of specific nutrients that can bias the intake of one food item over others [10]. A promising line of research aims to understand the cognitive challenges and feedback loops involved in integrating internal physiological needs with signals coding for macro- and micro-nutrient composition in foods [9]. External information can also be transmitted socially within insect populations [11], among physiologically differentiated members within social insect colonies [12,13] or among a diverse array of mutualist partners that convey their own physiological needs [14]. The social dimension in nutritional decisions is especially complex in social insects like ants (Hymenoptera: Formicidae), termites (Blattodea: Termitoidea), and bees (Hymenoptera: Apoidea) because foragers must harvest RNNs satisfying their own FNNs and those of their physiologically differentiated nestmates, often while simultaneously exchanging nutrients with unrelated mutualists.

## 3. Using Ants to Apply NG to Cognition Research

We propose that the existence of non-overlapping FNNs (and the distance between FNNs) of an individual insect, related kin or nestmates, and mutualist partners represent core cognitive challenges of eusocial life that can be profitably explored using NG techniques. NG is a multidimensional framework that allows one to study how insects prioritize nutritional tradeoffs when foraging for food items that are complex mixtures of macro- and micro-nutrients, minerals, vitamins, toxins, etc. A classic NG approach defines an insect’s intake target (IT) when foraging between two imbalanced nutritionally-defined diets in a choice laboratory experiment (Figure 2) [15]. Social insects likely integrate information about the nutritional needs of kin and mutualists via trophallaxis [16], cuticular hydrocarbons [17] or volatile organic compounds [18,19]. Based on such cues, the forager would likely select among foods to reach an optimal intake target (IT) while its nutritional interests would be split between provisioning itself (IT A), a relative (IT B), and a mutualist (IT C) (Figure 2A).

A nutrient-targeting hypothesis assumes that an insect dynamically navigates towards multiple specific ITs, expending time and energy to forage for the specific nutritional mixture to provision itself, a relative, or a mutualist depending on who has the most pressing nutritional needs (Figure 2A). Within social insects, nutrient-targeting may arise in small colonies with weakly defined division of labor, or that are specialized predators that only harvest one type of food (e.g., *Thaumatomyrmex*, [20]). In contrast, nutrient averaging may reflect neurophysiological constraints that limit tracking of dynamic nutritional signals, or the expectation that nutrients will be secondarily processed inside the nest by specialized castes [21]. We could expect nutrient-averaging to be more common in most ant colonies, where the RNNs harvested by any one ant will not perfectly match any individual’s FNN, but where the collectively harvested RNNs will almost certain be broad enough to target a range of specific FNN dimensions (Figure 2B).

NG now extends beyond studies of how and why foraging organisms prioritize multiple competing nutritional requirements [10] towards an integration with eco-evolutionary principles linking fundamental (FNN) and realized (RNN) nutritional niches to answer broad theoretical and applied ecological questions [22]. Researchers can first perform laboratory experiments defining an insect’s FNN by confining them to a broad range of nutritionally defined diets [23,24], and then visualize how performance (e.g., growth, survival, egg-laying rate) varies across this landscape (Figure 3A,B). The researcher can then perform field studies defining the insect’s RNN by measuring the nutritional composition of resources they actually harvest when foraging in nature (Figure 3C). By overlaying RNNs upon FNNs, the researcher can test whether and how insects nutritionally target specific FNN dimensions, and the extent to which they face nutritional tradeoffs between different measures of performance [24]. Moreover, while the framework we have discussed so far builds upon diets varying in two dimensions (e.g., protein, carbohydrates), recent NG approaches have been developed to map food mixtures in three dimensions (e.g., protein, carbohydrates, lipids) across right-mixture-triangle landscapes [24,25].

NG can also be used to understand how nutritional needs are expressed when cognitive processes linked to an individual’s FNN are modulated by its social context (e.g., nursing, parental care, courtship behavior, exchange with mutualists) [12,14,27]. For instance, an insect may adjust its IT to offset surplus nutritional subsidies from a mutualist, or it may harvest a different RNN when provisioning a mutualist than would be predicted from its own FNN dimensions. Below, we use a review of ant mutualisms to explore how NG can be used to identify the resulting cognitive challenges. Ants provide an ideal system for such inquiry since they are diverse (>15,000 species) and ecologically dominant [28], and since numerous individuals inside colonies communicate nutritional needs through trophallaxis [16] and pheromones [29]. Additionally, NG provided mechanistic insights into the longstanding observation that adult ant foragers often face challenges of provisioning developing brood with comparatively higher protein requirements [12,30], and that colonies shunt protein to brood while adult workers retain carbohydrates [31,32].

While most ant species scavenge diverse foods, NG has enabled researchers to rigorously explore the nutritional foraging rules underlying this food-level omnivory. Specifically, all ant species tested to date collectively harvest specific ITs in diet-choice experiments, and no-choice experiments have shown diverse nutritional strategies ranging from prioritizing strict regulation of either carbohydrates [27] or protein [14], to letting both nutrients fluctuate freely around the IT [24]. NG has also showed that a colony’s capacity for IT regulation can depend on the presence of developing brood [27], and that protein-biased diets can decrease both brood production and worker survival [27]. NG can also help visualize the distance between the FNNs of a foraging ant and those of its nestmates and mutualists—and thus highlights potential cognitive challenges in targeting adequate RNNs. For instance, while adult ants are predicted to have different FNNs than those of provisioned larval nestmates (Figure 4A), these FNNs are predicted to be more similar than those of unrelated mutualists that the ants must provision (e.g., fungal cultivars) (Figure 4C).

Ants also engage in diverse and ecologically important nutritional mutualisms to obtain stable access to limiting resources [33,34,35,36]. These mutualisms often reflect co-evolved relationships where one or both partners express adaptive traits whose evolution has been driven at least in part to meet the nutritional needs of its partner species, typically in exchange for some services (e.g., provisioned nutrients, protection against predators, dispersion, etc.) (Figure 4). When considering how mutualists signal their nutritional needs, we distinguish between ectosymbionts that live outside hosts and communicate via signals perceived by olfactory and gustatory chemosensory sensilla [16,17,18,19], and endosymbionts that live inside their hosts and communicate via signals perceived by internal receptors [37] (Figure 1).

Below, we briefly discuss the four general categories of ant mutualisms, focusing on how NG has been employed to understand their functioning. For more information on these mutualisms, see extensive reviews on bacteria [38,39], fungus [40,41], plants [33,42], and aphids [36]. We focus on mutualisms, while also noting that NG has been used to understand how parasites shape nutritional regulation in the ant colonies [43,44].

## 4. Mutualistic Relationships Involving Ants and an Unrelated Partner

### 4.1. Ant-Bacteria Mutualisms

Modern sequencing techniques have shown that bacteria played important roles in the radiation of ants across habitats and trophic levels over millions of years [35]. N-fixing gut bacteria have enabled ants to reach high numerical abundance in tropical forest canopies relative to prey availability [35,46] and thrive on protein-poor fungal diets in the fungus-farming attines [34,47]. Mutualists present in diverse ant species are capable of synthesizing urease to convert urea and uric acid into ammonium for amino-acid synthesis using the dietary nitrogen not metabolized by their host [45,48]. In this way, these ants could harness bacterial mutualists to upgrade toxic nitrogenous waste products and forage for FNNs with broader protein dimensions.

Identification of the exact nutrients exchanged between partners remains a challenge for ant-bacteria associations, and recent studies have partially achieved this through genetic sequencing [49]. For instance, the *Mollicutes* found within the workers of leafcutter ants are supplied with arginine by their ant hosts which is converted into ammonia, while the bacteria decompose citrate, derived from plant sap and fruit juice, into acetate for use in the host cell respiration [49]. Similarly, *Candidatus* Westeberhardia cardiocondylae within *Cardiocondyla* gut bacteriomes, retains the near complete metabolic pathway for the biosynthesis of 4-hydroxyphenylpyruvate, an important precursor for the essential amino acid tyrosine [50]. While such functional services provided by bacteria have been demonstrated in some ant-bacteria mutualisms, our understanding of how the bacteria and the nutrients they provide influence ant foraging decisions remains unclear. An ant’s gut bacteria may interfere with the host’s internal sensors responsible for evaluating the value of nutrients inside the gut, and in turn provide additional nutritional information to be processed by the insect host (Figure 1) [37,51]. The integration of this information would ultimately orientate an ant’s foraging decision toward specific source of nutrients that would fit both the bacteria needs as well as the need of the ant hosting the bacteria (Figure 4B) [52]. Through targeted in vitro cultivation of bacteria to determine bacteria FNN and direct manipulation of microflora within ants, NG could demonstrate that bacterial signals govern plasticity in an ant’s IT [45]. Moreover, nutrient restriction experiments affect gut microbial composition in mammals [53] and NG could provide evidence whether ants recover healthy microbial communities (following perturbation) by adjusting the RNNs they harvest.

### 4.2. Ant-Fungus Mutualisms

Ant-fungus mutualisms have evolved several times. For instance, the ant genera *Azteca*, *Allomerus*, *Petalomyrmex,* and *Aphomomyrmex* contain species that are mutualists with ascomycete fungi of the order *Chaetothyriales* [54,55]. At least four species of *Lasius* cover their ‘carton’ nests with a mycelial coat, with *L. fuliginosus* using a *Cladosporium*, and *L. umbratus* likely cultivating *Hormiscium pithyophilum* [56]. Using fluorescently dyed fungal hyphae [57] and stable isotope enrichment experiments (e.g., heavy forms of carbon, ^13^C and nitrogen, ^15^N) [58] the speed and targeted nature of nutrient exchange between ants and their fungal mutualists can be shown. This bidirectional nutritional mutualism is an ideal study system for the application of NG tools. For instance, NG can help visualize tradeoffs with specific nutrients involved in targeting the mutualist FNN, and cognitive challenges of provisioning an unrelated fungus whose FNN is likely more distant from the adult ants than those of developing larvae (Figure 4C).

The attine ants have evolved obligate fungiculture, with farming strategies of domesticated fungal crops radiating across 55–60 MYA [59]. Each farming colony cultivates a single fungal crop in monoculture [60] by harvesting resources ranging from detritus to fresh vegetation [61]. The ants convert these otherwise inedible substrates into nutritional mulches used to provision their fungi that in turn is consumed as their primary food source [34,58]. The *Atta* and *Acromyrmex* leafcutter ants display the highest degree of partner specialization, using mostly freshly cut vegetation to provision a *Leucoagaricus gongylophorus* cultivar that produces swollen hyphal tips called gongylidia that concentrate nutrients [62]. Gongylidia are consumed by ants in bundles called ‘staphylae’ and contain more carbohydrates and lipids than hyphae [63], with high levels of lipids (87% ergosterol) and carbohydrates (trehalose, mannitol, arabitol, and glucose), while also containing 18% proteins and free amino acids [64]. An NG in vitro approach was recently used to visualize the FNN of a fungal cultivar produced by the attine *Mycocepurus smithii*, and was coupled with whole colony NG no-choice experiments with nutritionally-defined diets showing how cultivar requirements for protein and carbohydrates impact and potentially constrain ant foraging decisions [14]. The ability of *M. smithii* to dynamically harvest RNNs that target their fungal cultivar’s FNN may require a unique suite of neurophysiological adaptations, enabling the ants to perceive fungal signals related to nutrients and toxins contained within harvested substrates. Attine ants may base their RNN provisioning decisions on primary or secondary metabolites (such as oxylipins, alkanes, acids, etc.) possibly produced by the fungus garden [65].

### 4.3. Ant-Plant Mutualisms

A diversification of plant mutualists has coincided with the diversification of ants over their evolutionary history [66]. Plants typically provide ants with extrafloral nectar [67,68] and/or specialized food bodies [69], as well as nesting sites within specialized structures in return for anti-herbivore defensive behaviors of the ants [42]. Plant rewards vary widely in nutritional composition, with extrafloral nectar being rich in sugars [68,70], and Beltian bodies of *Acacia* being composed of a relatively high concentration of proteins/amino acids (8–14%) and lipids (1–10%) [71]. Müllerian bodies of *Cecropia* are mainly composed of glycogen (39%) with few lipids (8%), while *Macaranga*-produced Beccarian bodies and the widely *Dicotylenodae*-produced pearl bodies are rich in lipids and protein [72,73], although pearl body nutrients also vary interspecifically [74]. Stable isotopes have shown that *Azteca* ants inhabiting *Cecropia* trees obtain ca. 18% of their carbon from their host plants [75], suggesting the mutualism merely supplements the ant’s diet. The plant hosts also receive nutrients from their resident ants, with up to 93% of a *Cecropia* plant’s nitrogen originating from debris deposited by *Azteca* [75]. Using NG, it should be possible to explore whether and how RNNs provided by this debris match a host’s plant FNN, and how this can shape the nutritional quality of the rewards the plants provide in return (Figure 4D).

Ants also rank among the most important insect seed dispersers with mutualistic seed-harvesting having convergently evolved across a wide range of ant taxa [76]. While ant-dispersed seeds can emit attractant volatiles [77], most studies have focused on the ‘elaiosomes’ that plants produce on their seeds [33]. Ants are attracted to the nutrients within elaiosomes, and carry the seeds back to their subterranean nests where they are protected from fire and herbivores [78,79]. Elaiosomes are typically rich in lipids, with species-specific blends of triglycerides, diglycerides, monoglycerides, free fatty acids, and also amino acids [80,81]. The high fatty acid concentration of mostly palmitic, palmitoleic, stearic, oleic and linoleic acids, as well as the diglyceride 1,2-diolein is more typical of insect prey than seeds [80,82]. This suggests that plant elaiosomes have evolved to coopt the pre-existing sensory traits of predatory ants for seed dispersal. Stable isotope labeling experiments have shown that elaiosome-derived nutrients are provisioned to ant larvae [83].

Some arboreal ant species plant the seeds of at least 15 epiphyte species to construct ‘ant gardens’ at the base of their carton nests [84,85]. The plants buttress the nest-structure and provide food for the ants, while in return gaining access to both ant defense and a nutrient-rich environment of ant-maintained soil rich in otherwise scarce nitrogen, phosphorus and potassium sources [86,87]. In general, ant species vary widely in the benefits they provide to plant hosts, and plants have thus evolved diverse chemical and physical partner-choice mechanisms [67,88]. It should be possible to infer whether a given ant species integrates information in partner-specific ways to adjust nutrient deposition and obtain specific rewards. NG can help visualize such partner filtering by showing whether host plants focus the RNN they provide to target the FNNs of a subset of ant species present in an arboreal ant community. Moreover, they may integrate signaling modalities into these nutritional mixtures to target pre-existing sensory mechanisms in potential ant mutualists [18].

### 4.4. Ant-Insect Mutualisms

Ant-insect relationships pose additional opportunities for NG inquiry since behaviors of both interacting species are mediated by cognition. The most common insect mutualists are herbivorous treehoppers, coccids and aphids [89], but also include other insects like *lycaenid* caterpillars [90]. These hemipterans generally extract plant phloem and excrete excess carbohydrates and essential amino acids that ants drink as liquid honeydew [91,92]. We focus below on ant-aphid mutualisms since these have been the focus of much of the nutritional research [92,93,94].

By feeding on honeydew, aphid-tending ants gain access to carbohydrates and amino acids that can act as a lifeline during colony foundation and subsequently drive colony growth [94,95]. In exchange, the ants defend aphids against disease [17], and predators (e.g., lacewings and ladybugs) [89]. The behavior of the ant *Lasius niger* is known to be directly influenced by the nutritional content of the aphid’s honeydew, with worker ants displaying a preference for trisaccharide-rich honeydews (e.g., melezitose and raffinose) [92,96]. In response, aphids can invest in the production of honeydew and adjust the production of the melezitose in the presence of the ants, increasing the rate of tending and the associated benefits, while paying an additional metabolic cost to produce these carbohydrates [93].

Using the logic of NG, variation in honeydew quality within tended aphid populations can represent a nutritional landscape upon which ants forage. For instance, honeydew asparagine and glutamine levels can vary with aphid age [97], and parasitized aphids often provide lower-quality honeydew [98,99]. Moreover, ants have been known to switch from tending aphids to eating them when the benefits of honeydew consumption are diminished [100]. Furthermore, some plants are hypothesized to provide alternative nutritional rewards to ants since aphids can reduce plant fitness [101]. The costs to aphids of providing suboptimal RNNs are thus high, and they likely use nutrients to reinforce ant tending behavior and inhibit predation. For instance, aphids may produce honeydew whose RNN dimensions target larval FNNs during periods of high colony growth, and adult worker FNNs during other periods of the colony’s life history (Figure 4E). If true, ants must determine whether the honeydew provided fits the current needs of the colony, and thus shift between tending and predation by integrating both nutritional quality and nestmates needs.

## 5. Conclusions

Ants provide model systems to study how social dynamics and mutualistic interactions shape the cognitive processes underlying nutritional foraging decisions. NG methods enable the visualization of how ants perceive, process, and interpret nutritional signals within a social context and how they adjust foraging relative to their FNN requirements and those of their related nestmates and unrelated mutualists. We propose that ants face increasing cognitive challenges when their FNNs have lower overlap with those of their mutualists. Since NG can accommodate the nutritional complexity of foods in N dimensions, it can be used to probe the key nutrients upon which the costs and benefits of engaging in a mutualism hinge, and can be applied to the four main classes of mutualisms in which ants engage (bacterial, fungal, plant, insect). This review thus provides a path towards new theory for the study of cognition and co-evolution.

## Figures and Tables

**Figure 1 insects-11-00053-f001:**
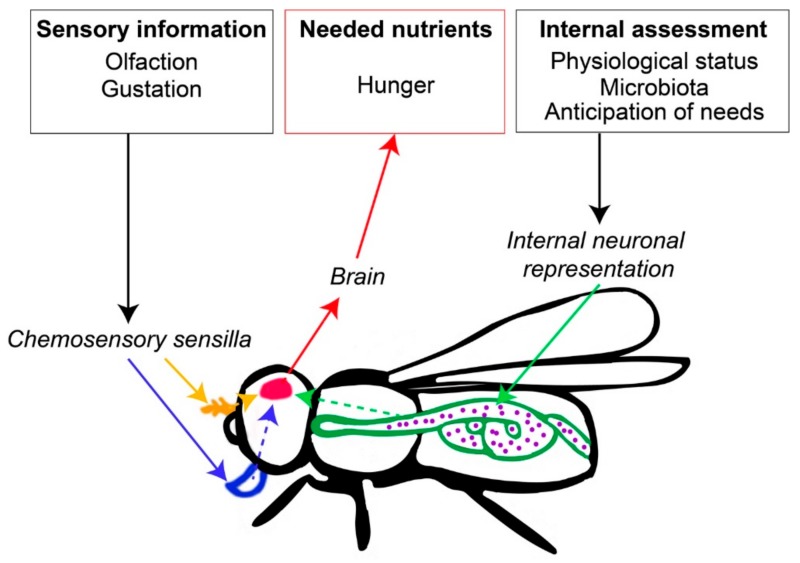
Basic neurophysiological processes involved in nutritional decision making. External sensory information is perceived by chemosensory sensilla involved in olfaction and gustation. Internal assessments of nutritional needs are detected by receptors and information is carried by neurons that relay physiological status, future needs and information from microbiota and from hemolymph. External and internal signals are transferred to the brain that integrates information to generate a hunger toward a specific nutrient. This information encodes the need for this nutrient.

**Figure 2 insects-11-00053-f002:**
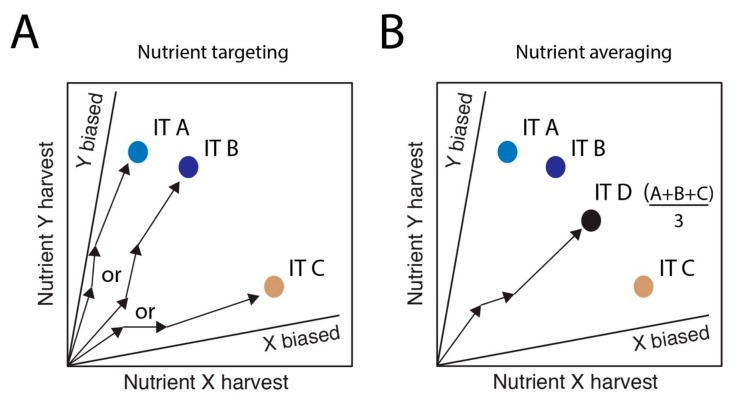
Nutritional geometry predictions of nutrient targeting and nutrient averaging in a diet-choice experimental set-up. (**A**) A nutrient-targeting hypothesis, assumes that a foraging insect can maintain information in separate neurophysiological channels about nutritional needs of itself, its nestmates and its mutualist(s). Here, an insect will forage across available foods in several distinct ways, targeting a diet mixture that specifically matches its own physiological needs (IT A), the needs of its nestmates (IT B), or targeting the specific nutritional deficiencies remaining after receiving nutrients from mutualists (IT C). The arrows represent the switches in foraging between X-biased and Y-biased diets used to reach the final IT. (**B**) A nutrient-averaging hypothesis assumes that the insect optimally integrates the different information from itself and partners to yield an averaged summary of communal needs. Here, an insect will target an intermediate diet mixture (IT D) that may save time and energy spent in foraging, but is also likely to suboptimally match the needs of each partner.

**Figure 3 insects-11-00053-f003:**
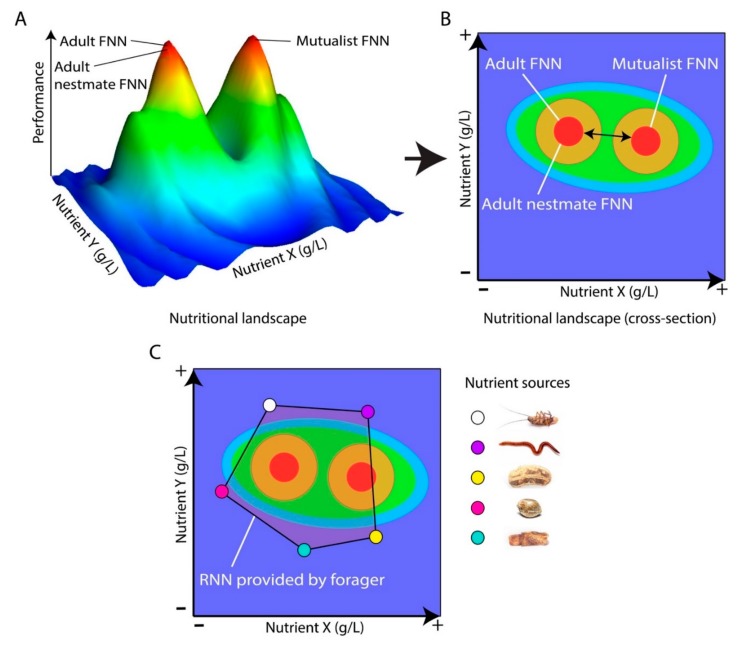
Visualizing the fundamental nutritional (FNN) and realized nutritional niche (RNN) of an individual insect, of its related kin (or nestmate), and unrelated mutualists on a nutritional landscape. (**A**) Nutritional landscape (adapted from the concept of fitness landscape [26]) allows one to visualize the FNN of an insect under controlled conditions when confined to a range of nutritionally defined diets and under a range of social conditions. For instance, the FNN of an individual is likely closer to those of its nestmates than those of an unrelated mutualist (e.g., fungal cultivar provisioned by leafcutter ants). Nutritional geometry (NG) is a useful tool for visualizing interactions of up to 3 co-limiting nutrients. (**B**) The impact of nutrients is usually investigated by combination of two or three (here two nutrients) and plotted into a cross-section (flattened representation) of the nutritional landscape that allows a simpler observation and interpretation of FNNs. (**C**) NG approaches can be taken to the field, so insects can be studied as they forage across real nutritional landscapes to select among many foods with different blends of nutrient X and Y. This yields the insect’s RNN harvested while being exposed to ecological pressures like competition or predation. By overlaying a field-RNN upon a lab-FNN, one can measure the degree of overlap, to assess whether and how nutrients harvested match the nutritional needs of the forager, nestmates and mutualist.

**Figure 4 insects-11-00053-f004:**
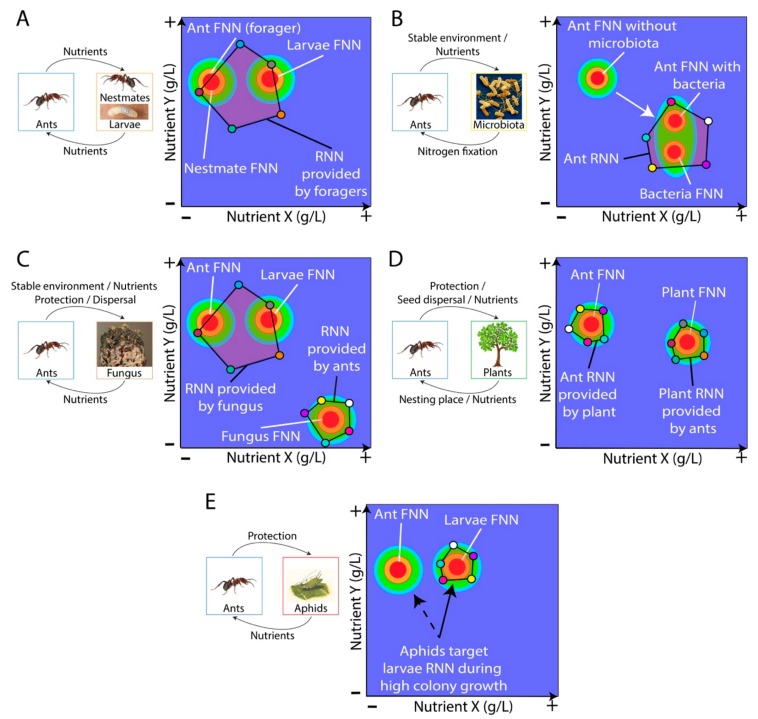
Predicting how ants navigate nutritional landscapes depending on their social environment. Light purple polygons represent RNNs harvested by foraging ants that are bounded by the nutrients contained in five foods shown in Figure 3C. (**A**) We predict adult ant nestmates have similar FNNs compared to more protein-biased FNNs of developing larvae. Despite these FNN differences, ants will collectively forage (i.e., nutrient averaging) to harvest a RNN that matches both adult and larval needs. (**B**) We predict that an ant’s FNN will change if it is experimentally deprived of its gut bacteria [45], and that a hypothetical bacterial species has its own FNN, which may influence the foraging behavior of its host ants closer to its own needs. (**C**) In the mutualism between attine ants and their domesticated fungal cultivars, ants provision their fungus with substrates harvested from the environment (e.g., fresh vegetation or detritus) and fungi assimilate these nutrients and concentrate them in fungal tissue consumed by the ants (gongylidia or hyphae). Nutrient exchange is thus bidirectional—ants provide their cultivars with a RNN and vice versa. We predict that provisioning a fungal cultivar—unique among ants to the attines—poses unappreciated nutrient-cognition challenges because its FNN is farther away from the adult ants than developing larvae. (**D**) Ants and plants exchange nutrients in diverse ways, reflecting the many evolutionary origins of these mutualisms. We generally predict that ants may provide a host plant with an RNN (e.g., through their nest) that should target a plant’s FNN so that it can maximize production of edible resources (e.g., Beltian bodies) whose RNN matches the ant’s FNN requirements. (**E**) Ants also exhibit diverse nutritional mutualisms with other insects. Here, we focus on aphids that provide ants with secreted honeydew, and predict that aphids can change the nutritional composition of honeydew to dynamically alter the RNN relative to the current requirements of the tending ant colonies. For instance, aphids may increase amino acid content in their honeydew to promote tending behaviors during periods when a growing colony is investing heavily in larvae with higher protein requirements.

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
