# Peer review of "Using Nutritional Geometry to Explore How Social Insects Navigate Nutritional Landscapes"

_insects, 2020, doi:10.3390/insects11010053_

Round 1

Reviewer 1 Report

This work reviews NG and incorporates its logic into the understanding of the cognitive challenges faced by the insects, and in particularly the ones that are both social and entangled in mutualistic symbiosis. The first sections are contextualizing the subject, and then the authors make an overview of NG predictions in terms of realized and fundamental nutritional niches using ants as model systems. The topic is no doubt relevant and up to date, and I consider the manuscript well structured, although here and then would profit from a more concise or shaped formulation (e.g. L30-33, L97-100). Even though I do not feel qualified to judge the use of the English language, I do feel that moderate English changes are required. In some cases, there is no concordance between the subject and the verb (e.g. L 211) or the preposition used does not seem correct (e.g. a on L194, about on L266).

On details, I would like to raise a point:

L72-75: It reads “The social dimension in nutritional decisions is especially complex in the ants (Hymenoptera: Formicidae) because foragers must harvest RNNs satisfying their own FNNs and those of their physiologically differentiated nestmates, often while simultaneously exchanging nutrients with unrelated mutualists.”

I would like to suggest not to reduce the message to ants, but to reformulate the sentence in order to accommodate other social insects that have similar colony structures and are also known by harbouring or interacting with unrelated mutualists (e.g. termites, bees). Later, the review narrows in into ants, arguing their suitability as “model” organisms for studying NG in its different axis, so I reckon here it would make sense to keep it broader.

And I would like to highlight some sentences where I had more difficulties in understanding the message and likely need some rephrasing:

L30-L31: It reads “can be most adequately understood in the context of nutrients these mutualists provided and consume” should it read “can be most adequately understood in the context of nutrients THAT these mutualists provided and consume”?

L63-L64: It reads “underlying regulatory processes (reviewed in [9] and references within; [10]).” Because [9] is a review and clearly stated as such, I see no need to add ‘and references within’.

L94-L100: When you state “We could expect this strategy to be …” It is not clear to what strategy it refers to: a nutrient-targeting hypothesis? as opposing to nutrient-averaging? I do get the idea, but I would suggest some rephrasing for clarification.

L194: It reads “change if it is experimentally deprived of a its gut bacteria”; the ‘a’ should be removed

L211-213: It reads “The advent of 16S rRNA sequencing techniques allowing for the identification of the microbes have shown that bacteria played important roles…”; if you are referring to the advent of 16S rRNA sequencing techniques, I think it should read … has shown that… However, I think that the main subject of the sentence should be “The identification of the microbes, which was made possible by the advent of 16S rRNA sequencing techniques, has shown that….”. Either way, the sentence needs rephrasing.

Author Response

To the editors and reviewers of Insects,

We would like to thank the two anonymous reviewers who have examined our manuscript and provided comments. Please find a revised version of the manuscript as well as a point-by-point answer for the different comments. Besides integrating reviewer’s comments, we provided a more appropriate title, carefully corrected language errors and improved written style to facilitate reading.   

Reviewer 1:

This work reviews NG and incorporates its logic into the understanding of the cognitive challenges faced by the insects, and in particularly the ones that are both social and entangled in mutualistic symbiosis. The first sections are contextualizing the subject, and then the authors make an overview of NG predictions in terms of realized and fundamental nutritional niches using ants as model systems. The topic is no doubt relevant and up to date, and I consider the manuscript well structured, although here and then would profit from a more concise or shaped formulation (e.g. L30-33, L97-100). Even though I do not feel qualified to judge the use of the English language, I do feel that moderate English changes are required. In some cases, there is no concordance between the subject and the verb (e.g. L 211) or the preposition used does not seem correct (e.g. a on L194, about on L266).

à We rephrased L30-33: Now ‘In this review, we focus on ants and use the logic of nutritional geometry (NG) to explore: 1) the cognitive traits that insects need to navigate complex nutritional landscapes, and how 2) social environments and 3) mutualist-derived nutrients mediate these foraging decisions.’ We rephrased L97-100 (L160-161 with track changes in Word): ‘Within social insects, nutrient-targeting may arise in small colonies with weakly defined division of labor, or that are specialized predators that only harvest one type of food’. We also edited L194, L211, L266, and corrected typos throughout the manuscript.

On details, I would like to raise a point:

L72-75: It reads “The social dimension in nutritional decisions is especially complex in the ants (Hymenoptera: Formicidae) because foragers must harvest RNNs satisfying their own FNNs and those of their physiologically differentiated nestmates, often while simultaneously exchanging nutrients with unrelated mutualists.” I would like to suggest not to reduce the message to ants, but to reformulate the sentence in order to accommodate other social insects that have similar colony structures and are also known by harbouring or interacting with unrelated mutualists (e.g. termites, bees). Later, the review narrows in into ants, arguing their suitability as “model” organisms for studying NG in its different axis, so I reckon here it would make sense to keep it broader.

à We rephrased the sentence that now includes termites and bees. ‘The social dimension in nutritional decisions is especially complex in social insects like ants (Hymenoptera: Formicidae), termites (Blattodea: Termitoidea), and bees (Hymenoptera: Apoidea) because foragers must harvest RNNs satisfying their own FNNs and those of their physiologically differentiated nestmates, often while simultaneously exchanging nutrients with unrelated mutualists.’ on L120-124 (with track changes in Word).

And I would like to highlight some sentences where I had more difficulties in understanding the message and likely need some rephrasing:

L30-L31: It reads “can be most adequately understood in the context of nutrients these mutualists provided and consume” should it read “can be most adequately understood in the context of nutrients THAT these mutualists provided and consume”?

à We modified the sentence for ‘Insects also often engage in diverse nutritional mutualisms with bacteria, fungi, plants, and other insects, and we propose that these nutritional subsides can complicate the cognitive challenges of harvesting an optimal RNN.’ on L28-30 (with track changes in Word).

L63-L64: It reads “underlying regulatory processes (reviewed in [9] and references within; [10]).” Because [9] is a review and clearly stated as such, I see no need to add ‘and references within’.

à We have deleted ‘and references within’.

L94-L100: When you state “We could expect this strategy to be …” It is not clear to what strategy it refers to: a nutrient-targeting hypothesis? as opposing to nutrient-averaging? I do get the idea, but I would suggest some rephrasing for clarification.

à We clarified the sentence according to both Reviewer 1 and 2. The sentence is now ‘Within social insects, nutrient-targeting may arise in small colonies with weakly defined division of labor, or that are specialized predators that only harvest one type of food’ on L160-161 (with track changes in Word).

L194: It reads “change if it is experimentally deprived of a its gut bacteria”; the ‘a’ should be removed

à We deleted the ‘a’.

L211-213: It reads “The advent of 16S rRNA sequencing techniques allowing for the identification of the microbes have shown that bacteria played important roles…”; if you are referring to the advent of 16S rRNA sequencing techniques, I think it should read … has shown that… However, I think that the main subject of the sentence should be “The identification of the microbes, which was made possible by the advent of 16S rRNA sequencing techniques, has shown that….”. Either way, the sentence needs rephrasing.

à We rephrased the sentence. Now it reads “Modern sequencing techniques have shown that bacteria played important roles in the radiation of ants across habitats and trophic levels over millions of years” on L424-425 (with track changes in Word).

Thank you for considering the modification with provided to the manuscript. We look forward to hearing from you.

Sincerely,

Dr. Antonin Crumière, other coauthors and affiliations

Reviewer 2 Report

When I read the abstract, I was waiting for another paper of broken symmetries on paths. I did not know the field of NG and it has been a real discovery and pleasure at the same time. But, from my perspective in collective intelligence and ecology of mutual communities, it has been hard for me to follow the explanations of the paper. If I understood correctly, the main purpose of the paper is to expose the cognitive tension between the individual foraging strategy and the necessity of feeding of the nest-mates and the mutualistic elements into the individual and into the nest. The authors propose that NG in combination with the field of eco-evolutionary knowledge can be an useful approach to understand this tension. Although the problem of the development of collective cognition from individual elements is a very interesting challenge not solved yet, the authors propose an interesting (but sometimes oversimplified) approach with which to explore this problem through the NG and eco-evolutionary theories.
I have some concerns with definitions. Starting with the main concept on the paper, cognition, It is clear that the social insect is a cognitively capable individual, but these individual capabilities are relevant on the emergence of the entire colony cognition. I consider that a key aspect in the emergence of this collective cognition is communication, and some relevant nutritional skills are based on the collective cognition of the nest. Eusocial insects display a huge diversification of cuticular pheromones used to convey multiple signals unique to colony life. Another famous example is the honeybee waggle dance. While solitary insects may have the motivation to conceal a newly found item for personal consumption, the bees have evolved an elaborate communication scheme which allows them to share this location. I consider that a further exploration of how the individual perception of nutritional requirements behaves in a collective interacting system could bring light to this problem.
I have another problem with the definition of RNN and FNN, according to ecology, fundamental niche is the entire set of conditions under which an animal (population, species) can survive and reproduce itself, while realized niche is the set of conditions actually used by a given animal (pop, species), after interactions with other species (predation and especially competition) have been taken into account. Ergo, the realized is the effectively used fraction of the fundamental. The maximal performance of an animal can be achieved with a nutritional intake that is a fraction of the fundamental niche. In fact, the ecological range of habitats of a species is based on that. Individuals can maximize performance even with a tiny fraction of the fundamental niche. Unless it is a highly specialized species. The definition of RNN and FNN leads me to confusion.

Some precise concerns:
94 A nutrient-targeting hypothesis assumes that this insect dynamically navigates towards multiple specific ITs, expending time and energy to forage for the specific nutritional mixture to provision itself, a relative, or a mutualist depending on who has the most pressing nutritional needs. How are these specific needs of the nest-mate and the mutualistic organism perceived by the individual insect?
236 Sometimes looks like insects enjoy super-cognition to test own nutritional needs and those around him, food composition and modulate its own microbiome community. How can causality be established?
336 It is not clear for me, after read the paper, how social dynamics and mutualistic interactions shape the cognitive processes underlying nutritional foraging decisions. I consider that your conclusions are not sufficiently based. I consider that causality can not be established just considering NG and RNN/FNN overlap. 

Author Response

To the editors and reviewers of Insects,

We would like to thank the two anonymous reviewers who have examined our manuscript and provided comments. Please find a revised version of the manuscript as well as a point-by-point answer for the different comments. Besides integrating reviewer’s comments, we provided a more appropriate title, carefully corrected language errors and improved written style to facilitate reading.   

Reviewer 2:

When I read the abstract, I was waiting for another paper of broken symmetries on paths. I did not know the field of NG and it has been a real discovery and pleasure at the same time. But, from my perspective in collective intelligence and ecology of mutual communities, it has been hard for me to follow the explanations of the paper. If I understood correctly, the main purpose of the paper is to expose the cognitive tension between the individual foraging strategy and the necessity of feeding of the nest-mates and the mutualistic elements into the individual and into the nest. The authors propose that NG in combination with the field of eco-evolutionary knowledge can be an useful approach to understand this tension. Although the problem of the development of collective cognition from individual elements is a very interesting challenge not solved yet, the authors propose an interesting (but sometimes oversimplified) approach with which to explore this problem through the NG and eco-evolutionary theories. 

I have some concerns with definitions. Starting with the main concept on the paper, cognition, It is clear that the social insect is a cognitively capable individual, but these individual capabilities are relevant on the emergence of the entire colony cognition. I consider that a key aspect in the emergence of this collective cognition is communication, and some relevant nutritional skills are based on the collective cognition of the nest. Eusocial insects display a huge diversification of cuticular pheromones used to convey multiple signals unique to colony life. Another famous example is the honeybee waggle dance. While solitary insects may have the motivation to conceal a newly found item for personal consumption, the bees have evolved an elaborate communication scheme which allows them to share this location. I consider that a further exploration of how the individual perception of nutritional requirements behaves in a collective interacting system could bring light to this problem. 

à We have developed the paragraph on L152-164 (with track changes in Word) to explain how an individual can receive information from partners and how the quantity of information from the social context can impact the foraging activity. We also developed the communication aspect of ants on L283-285 (with track changes in Word)  ‘Ants provide an ideal system for such inquiry since they are diverse (>15,000 species) and ecologically dominant [28], and since numerous individuals inside colonies communicate nutritional needs through trophallaxis [16] and pheromones [29].’ We mentioned the mechanisms associated with food detection in introduction but we consider further discussion of this point is out of the scope of our review. 

I have another problem with the definition of RNN and FNN, according to ecology, fundamental niche is the entire set of conditions under which an animal (population, species) can survive and reproduce itself, while realized niche is the set of conditions actually used by a given animal (pop, species), after interactions with other species (predation and especially competition) have been taken into account. Ergo, the realized is the effectively used fraction of the fundamental. The maximal performance of an animal can be achieved with a nutritional intake that is a fraction of the fundamental niche. In fact, the ecological range of habitats of a species is based on that. Individuals can maximize performance even with a tiny fraction of the fundamental niche. Unless it is a highly specialized species. The definition of RNN and FNN leads me to confusion.

à We agree and have edited the definitions of FNN and RNN on L25-28 accordingly. The new sentence is ‘A central challenge is to harvest a realized nutritional niche (RNN) based on available resources (e.g. in the presence of predation and competition) that optimally targets fitness-maximizing dimensions of the innate fundamental nutritional niche (FNN).’.

Some precise concerns: 94 A nutrient-targeting hypothesis assumes that this insect dynamically navigates towards multiple specific ITs, expending time and energy to forage for the specific nutritional mixture to provision itself, a relative, or a mutualist depending on who has the most pressing nutritional needs. How are these specific needs of the nest-mate and the mutualistic organism perceived by the individual insect?

à This aspect was discussed later in the manuscript. We now rephrased L152-156 (with track changes in Word) to include this information earlier. The new sentence is ‘Social insects likely integrate information about the nutritional needs of kin and mutualists via trophallaxis [16], cuticular hydrocarbons [17] or volatile organic compounds [18, 19]. Based on such cues, the forager would likely select among foods to reach an optimal intake target (IT) while its nutritional interests would be split between provisioning itself (IT A), a relative (IT B), and a mutualist (IT C) (Figure 2A).’

236 Sometimes looks like insects enjoy super-cognition to test own nutritional needs and those around him, food composition and modulate its own microbiome community. How can causality be established?

à We have rephrased and developed this sentence to explain more clearly the logic. The sentence is now ‘Nutrient restriction experiments affect gut microbial composition in mammals [53] and NG could provide evidence whether ants recover healthy microbial communities (following perturbation) by adjusting the RNNs they harvest’ on L448-450 (with track changes in Word).

336 It is not clear for me, after read the paper, how social dynamics and mutualistic interactions shape the cognitive processes underlying nutritional foraging decisions. I consider that your conclusions are not sufficiently based. I consider that causality can not be established just considering NG and RNN/FNN overlap. 

à The connection with the cognition hypothesis was missing here. Our goal is to frame new hypotheses about how mutualisms function and how they are evolutionary stable. The ability of NG to ‘establishing causality’ will vary across systems and the degree to which NG is combined with our sources of inference.  We have edited this section accordingly on L657-659 (with track changes in Word) ‘If true, ants must determine whether the honeydew provided fits the current needs of the colony, and thus shift between tending and predation by integrating both nutritional quality and nestmates needs.’.

Thank you for considering the modification with provided to the manuscript. We look forward to hearing from you.

Sincerely,

Dr. Antonin Crumière, other coauthors and affiliations
